# Think Before You Act: Unified Policy for Interleaving Language Reasoning with Actions

**Lina Mezghani**
Meta AI & Inria*

**Piotr Bojanowski**
Meta AI

**Karteek Alahari**
Inria*

**Sainbayar Sukhbaatar**
Meta AI

## Abstract

The success of transformer models trained with a language modeling objective brings a promising opportunity to the reinforcement learning framework. Decision Transformer (Chen et al., 2021) is a step towards this direction, showing how to train transformers with a similar next-step prediction objective on offline data. Another important development in this area is the recent emergence of large-scale datasets collected from the internet, such as the ones composed of tutorial videos with captions where people talk about what they are doing. To take advantage of this language component, we propose a novel method for unifying language reasoning with actions in a single policy. Specifically, we augment a transformer policy with word outputs, so it can generate textual captions interleaved with actions. When tested on the most challenging task in BabyAI, with captions describing next subgoals, our reasoning policy consistently outperforms the caption-free baseline.

## 1 Introduction

Transformer-based (Vaswani et al., 2017) large language models (LLMs) trained on a massive amount of text data have shown impressive results in various language understanding tasks (Brown et al., 2020). Their application also goes beyond language domains, ranging from instruction following (Hill et al., 2020) to vision-language navigation (Majumdar et al., 2020). This progress suggests that LLMs are powerful not only for purely linguistic modeling, but also for setups that require planning and sequential reasoning. Interestingly, Li et al. (2022) show that this even holds for tasks that do not involve actual language at all, but only sequential string tokens, which highlights the compositional power of these models.

Along with the rise of LLMs, the field of reinforcement learning (RL) has also been impacted by advances in sequential decision-making. Indeed, the development of offline RL (Levine et al., 2020), that is, exploiting pre-collected datasets to learn policies without interacting with the environment, has enabled a new way of learning controllable agents—from the sequence modeling perspective (Janner et al., 2021). In this light, Decision Transformer (Chen et al., 2021) shows how to learn a policy using a causal transformer (Vaswani et al., 2017), and follow-up works (Putterman et al., 2021; Correia & Alexandre, 2022) propose ways of training them without rewards, in a goal-conditioned formulation.

Replicating the success of LLMs in RL also requires large-scale training data, such as Minedojo (Fan et al., 2022) that gathers, among other things, a collection of YouTube video tutorials on MineCraft. These videos contain captions that often explain what the human is doing while playing, creating an exciting opportunity to leverage language-based reasoning with decision making in RL.

In this paper, we propose a method for training an agent that is capable of interlacing language reasoning with performing actions in an environment. This allows us to fully leverage captions in offline training data, in addition to actions and observations. Our method relies on a unified policy that is capable of choosing between thinking verbally and acting. We achieve this by equipping

---

*†Univ. Grenoble Alpes, Inria, CNRS, Grenoble INP, LJK, 38000 Grenoble, France

Reincarnating Reinforcement Learning Workshop at ICLR 2023

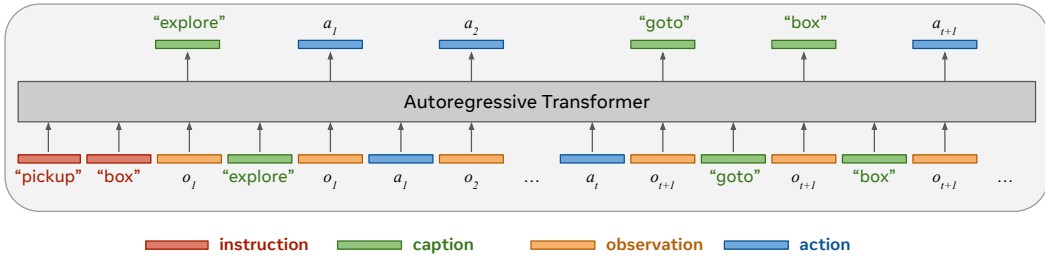

Figure 1: Given an instruction, our transformer policy can generate language based reasoning tokens interleaved with sequence of actions in the environment.

the policy with additional word outputs so it can decide when to act and when to perform language reasoning. To this end, we train an auto-regressive transformer, conditioned on the instruction and observations, to predict both actions and text captions in a unified way. At test time, given a language instruction, our model predicts actions towards the goal, as well as text tokens to reason, as shown in Fig. 1, where the captions define which subgoals it has to reach to solve the task.

In order to study this problem in a controlled setting, we experiment on BabyAI (Chevalier-Boisvert et al., 2018), a grid world platform to study language-grounded tasks. Specifically, we focus on language reasoning that lays out the next subgoal. To achieve this, we first generated a dataset of trajectories with the expert bot provided by Chevalier-Boisvert et al. (2018). We then leveraged the underlying subgoals used by the bot to create text descriptions aligned with the trajectories, *i.e.*, captions. We evaluate the performance of our model zero-shot, on language-specified instructions, and study its generalization to unseen maze configurations on several setups, including the BossLevel, the most difficult task of BabyAI. We show that leveraging text captions greatly improves the performance compared to the caption-free baseline, particularly for harder tasks that require planning and long-term reasoning.

In summary, we make the following contributions: **(i)** we propose an algorithm for generating text-augmented expert trajectories on BabyAI (Chevalier-Boisvert et al., 2018) that constitutes a toy environment for emulating tutorial videos, **(ii)** we present an auto-regressive transformer architecture to learn to predict both actions and captions in a unified way, and **(iii)** we show that it outperforms the caption-free baseline, particularly on tasks that involve long-term sequential planning.

## 2   Related Work

**Offline datasets in RL.** The data collection technique is an important aspect when studying the training of policies from pre-collected datasets. In this context, prior works assumed access to policies trained by task-specific rewards (Fu et al., 2020; Gulcehre et al., 2020) or proposed to leverage unsupervised exploration policies (Eysenbach et al., 2018; Yarats et al., 2021) to collect datasets for offline RL (Yarats et al., 2022; Lambert et al., 2022). These works are often limited to learning a restricted set of tasks, specified with hand-crafted reward functions. More recently, the accessibility of large-scale datasets (Baker et al., 2022; Fan et al., 2022) from the internet enabled new possibilities for learning offline policies with language-based task specification. Indeed, these works propose to exploit a dataset of YouTube videos, that are augmented with text sources at several stages, including captions throughout the video. However, this data source has one crucial drawback, as the actions that the player is executing (*i.e.*, the keyboard buttons or the mouse movements) are missing, and consequently, the videos cannot be considered as *expert trajectories*. To overcome this issue, Baker et al. (2022) propose to collect a small dataset of trajectories annotated by human experts. Using this data, the authors propose to train an inverse dynamics model (Nguyen-Tuong et al., 2008) to predict the actions, given past and future frames, and use it to annotate the large dataset of YouTube videos with corresponding actions.

**Transformers in RL.**   Using Transformers in sequential decision-making (Li et al., 2023) was first proposed by Janner et al. (2021); Chen et al. (2021). The latter present decision transformer, a causal transformer model with masked attention layer used to train policies on offline datasets.

One particularity of this model is that it uses returns-to-go, that is, discounted cumulated rewards, to derive optimal behaviour from the training data. By leveraging these cumulative rewards, the model learns the interesting transitions, even when the offline dataset is not collected with an expert policy. The drawback of this method, however, is that the initial return-to-go value must be arbitrarily chosen at test time. To avoid relying on these, Correia & Alexandre (2022) propose a hierarchical formulation of the decision transformer, where a high-level policy outputs subgoals for the low-level policy. Alternatively, Putterman et al. (2021) propose a goal-conditioned version of the decision transformer, where demonstrations are preceded with a language instruction describing the goal of the trajectory. In this setup, trajectories must be collected by expert policies since the transformer only imitates the demonstrations present in the dataset. Related to this line of work, our model leverages not only language instructions, but also text description along the trajectory, for effective learning and better generalization.

**Language-conditioned Offline RL.** Advances in both offline RL and transformer models for sequential decision-making have led to an interesting research direction, that aims at taking advantage of text data for offline RL. Several works showed promising results on datasets as large as MineDojo (Fan et al., 2022) by designing specific reward functions on top of the offline data, either manually for a specific task like crafting diamonds (Baker et al., 2022), or by learning the alignment between text and images in the video (Fan et al., 2022), in a CLIP (Radford et al., 2021) fashion. A very recent line of study proposed to use large language models on offline datasets for decision-making tasks: Zhang et al. (2022) show that task instructions can be effectively used to pre-train offline policies and Carta et al. (2023) train an adaptive LLM-based policy in language-grounded tasks. Closely related to our work, Li et al. (2022) show the effectiveness of LLMs on decision-making tasks, specifically on the BabyAI environment (Chevalier-Boisvert et al., 2018). Their idea is to convert actions and observations to text input, and fine-tune a pre-trained GPT-2 (Radford et al., 2019) language model on a set of expert trajectories. Apart from the observation conversion part, our model is trained on the same type of data as this work, and our goal is to show the importance of using subgoal descriptions for the stability and performance of training.

## 3 Problem Formulation

We consider a goal-conditioned partially observable Markov decision process (POMDP), defined by a set of states, actions, observations, goals and a transition model, that maps the current state and action to the next state. In a POMDP, the observation $o_t$ at timestep $t$ only captures a portion of the underlying state, and an optimal policy must take as input not only the last observation $o_t$, but also a history of previous observations and actions (Parr & Russell, 1995) $h_t = (o_0, a_0, ..., o_{t-1}, a_{t-1})$. Our goal is to learn a goal-conditioned policy $\pi(a_t|m, h_t, o_t)$ which, given an instruction (or mission) $m$, the history of previous observations and actions $h_t$, and last observation $o_t$, outputs a probability over the set of actions.

In this work, we assume that we do not have access to the POMDP at train time, but instead, that the policy must be trained offline, on a dataset of pre-collected expert trajectories $\mathcal{D}$. This dataset contains trajectories $(m, (o_t, a_t)_{0 \leq t < T})$ where $m$ is the goal of the trajectory specified with natural language (*i.e.*, an instruction), such that the sequence $(o_t, a_t)_{0 \leq t < T}$ satisfies goal $m$. We also assume that the training trajectories are augmented with natural language captions $c_t$ at every timestep, that is, trajectories are of the form: $(m, (o_t, a_t, c_t)_{0 \leq t < T})$. However, these captions are not available during the evaluation stage. The goal is therefore to learn a policy, which at test time, is given a instruction, and must perform the actions that satisfy the goal described by the instruction.

## 4 Method

We will now present our approach, which relies on the idea that language annotations, e.g., captions from YouTube gaming tutorials, provide useful information about which subgoals the player is solving while playing. In MineCraft for instance, in a tutorial on how to craft diamonds, the video will explain the different steps that should be executed to achieve the goal, including crafting a table and a pickaxe. Since captions are not available at test time, when the agent is deployed in the environment, we propose to train an auto-regressive transformer on the offline dataset that, given the instruction and sequence of observations, will not only learn to predict actions, but also captions.

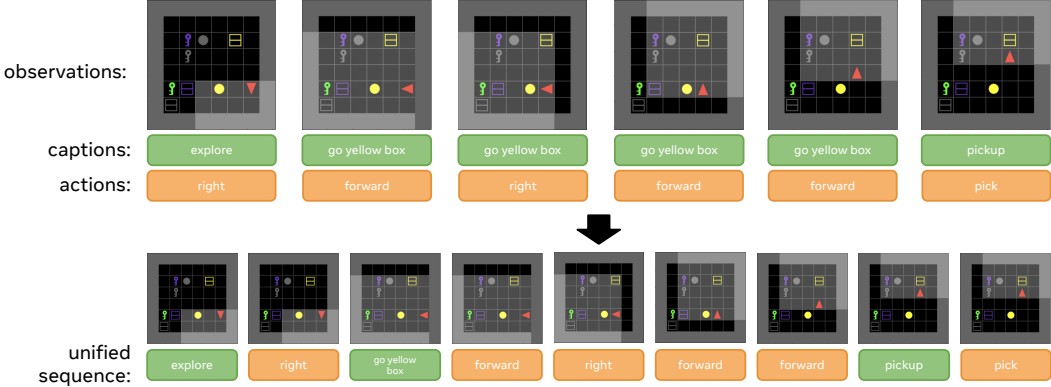

Figure 2: Visualisation of the trajectory to sequence process. The top row shows the original trajectory, with observations, actions and captions, and the bottom row shows how observations are duplicated, and how action and captions organized in the unified sequence.

## 4.1 Unifying actions and language reasoning

In the formalism described in Sec. 3, we assume that captions and actions have the same temporal granularity, *i.e.*, that a new caption appears exactly when a new action is executed. In practice however, in tutorial videos, the granularity of action and caption sequences might be different, as same captions can last for several steps, or conversely, a long caption can occur while no new action appeared.

In our approach, we propose to unify actions and language reasoning in a joint sequence, and train a model that can predict both modalities. We therefore create a new sequence $(s_i, x_i)_{0 \leq i < N}$ from the $\mathcal{T} = (o_t, a_t, c_t)_{0 \leq t < T}$ trajectory. The newly created sequence includes a list of tokens $(x_i)_{0 \leq i < N}$ that can be either caption tokens or actions, as well as a list $(s_i)_{0 \leq i < N}$ of observations, obtained by duplicating elements in $(o_t)_{0 \leq t < T}$. The general idea is

---

**Algorithm 1** Sequence from trajectory

**Input:** trajectory $\mathcal{T} = (o_t, a_t, c_t)_{0 \leq t < T}$
$s_0, x_0 := o_0, c_0$
$s_1, x_1 := o_0, a_0$
$i := 1$
**for** $t = 1$ **to** $T - 1$ **do**
  **if** $c_t \neq c_{t-1}$ **then**
    $i := i + 1$
    $s_i, x_i := o_t, c_t$
  **end if**
  $i := i + 1$
  $s_i, x_i := o_t, a_t$
**end for**

---

to go through the trajectory $\mathcal{T}$, and whenever a new caption is encountered, insert its tokens in the sequence by duplicating the corresponding observation. In practice, we convert actions to natural language, as done in prior work (Li et al., 2022; Carta et al., 2023), so that actions and captions map to the same vocabulary. The process is described in Alg. 1, and an illustrated example is shown in Fig. 2. The dataset of trajectories $\mathcal{D}$ is therefore transformed into a dataset $\mathcal{B}$ of unified action-caption sequences of the form $(m, (s_i, x_i)_{0 \leq i < N})$.

## 4.2 Auto-regressive transformer for generating both language and actions

We train an auto-regressive transformer on the set of sequences $\mathcal{B}$, as shown in Fig. 1. The model first encodes the instruction sentence $m$ by tokenizing it into a sequence of length $m = (m_0, ..., m_{n-1})$, and embedding every token. For computational efficiency, we sample sub-sequences $(s_i, x_i)_{t \leq i < t+K}$ of length $K$ from the full sequences in $\mathcal{B}$. We pass the observations $(s_i)_{t \leq i < t+K}$ through a convolutional network, and embed the action or caption tokens $(x_i)_{t \leq i < t+K}$ before passing them to the transformer model. The model predicts the action or caption token at every step in the trajectory in an auto-regressive manner, and is trained to minimize the token prediction error across the trajectory using the cross-entropy loss.

Similar to Vaswani et al. (2017), each element in the sequence is summed with a positional encoding, to make sure that the model takes advantage of the order of the sequence. While instruction tokens

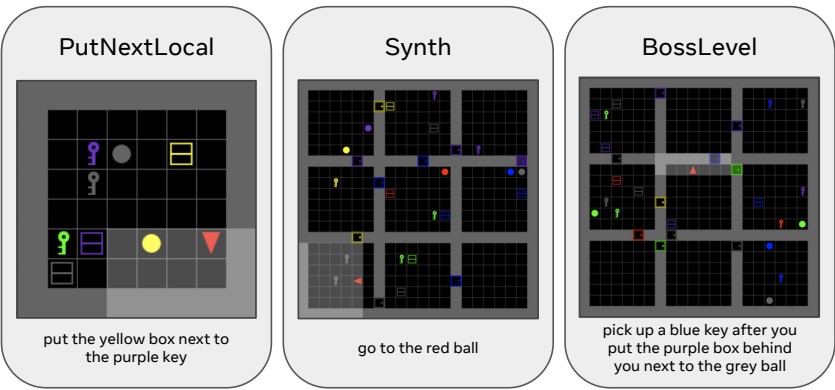

Figure 3: The three levels from BabyAI that we consider in this work, along with an example of observation and instruction for each of these environments.

$m_j$ are summed with encodings of $j$ indicating their absolute positions, $s_i$ and $x_i$ tokens are summed with the encoding of $i$ informing their position in the sequence. Note that this is different from timestep encodings because $i$ is incremented by both caption tokens and environmental steps.

At test time when the agent has to act in the environment, we sequentially generate tokens from the transformer model by feeding the current observation. If the generated token belongs to the caption vocabulary, we simply feed it back to the model and continue the generation similar to language models. Only when the model generates an action token, we perform this action in the environment and feed the updated observation to the model. This way the model itself decides when to reason and when to act, interleaving multiple reasoning captions throughout an episode.

## 5 Experiments

### 5.1 Data generation in BabyAI

We generate a dataset of expert trajectories aligned with natural language captions in the BabyAI (Chevalier-Boisvert et al., 2018) environment. BabyAI is a grid world platform developed to train agents grounded with natural language instructions. The platform contains several levels that vary in difficulty, with different types of instructions and distractors. In this paper, we consider the following three levels, of increasing difficulty:

- **PutNextLocal**: the agent must put an object next to another one, with distractors.

- **Synth**: the agent must perform one of the four possible instructions **(i)** *open* a door specified by its color, **(ii)** *go to* an object specified by its nature and color, **(iii)** *pick up* an object specified by its nature and color, and **(iv)** *put* an object next to another one.

- **BossLevel**: the hardest task of the BabyAI suite. The mission is a combination of tasks from the **Synth** level, where objects can be specified by their type and color, but also by their location on the map (*e.g.*, "the door on your left" or "the ball behind you"). Moreover, the task combination might have to be realised in a particular order.

An example of an environment and text instruction for each level is shown in Fig. 3. In all three environments, the observations cover the $7 \times 7$ area visible to the agent, and is given as a tensor of size $7 \times 7 \times 3$, where the last dimension describes the type of object present at the corresponding location, its color and its state (*e.g.*, for doors, either *open*, *locked* or *closed*).

Together with the platform, Chevalier-Boisvert et al. (2018) also provide an expert bot, that can solve any language-grounded instruction in the environment. It is implemented by setting subgoals to itself and solving them sequentially. We therefore used this bot to collect expert trajectories, and processed the underlying subgoals used by the bot to generate natural language descriptions of the sub-tasks that the agent is achieving.

Table 1: Success rate (mean and standard error) on the three BabyAI levels with different amounts of training data. We show results for our method, as well as two caption-free baselines, including **BabyAI-Ori-1M**, the imitation learning baseline from the original BabyAI paper.

| | PutNextLocal-100k | Synth-500k | BossLevel-1M |
|---|---|---|---|
| **BabyAI-Ori-1M** | 99.2 | 97.3 | 77.0 |
| **Baseline** | **99.5** $_{\pm 0.2}$ | **97.9** $_{\pm 0.2}$ | 73.6 $_{\pm 8.0}$ |
| **Our Method** | **99.6** $_{\pm 0.1}$ | 96.4 $_{\pm 0.3}$ | **85.2** $_{\pm 0.5}$ |

Note that for each trajectory, the environment configuration is different, *i.e.*, the location of doors, objects, as well as the initial agent location if randomized. Moreover, the seed for generating evaluation episodes is distinct from the train one, which means that environment configurations from the evaluation set are not seen during training. We thus generate datasets of varying size, ranging from 50k to 1M trajectories, and evaluate all the models on a fixed set of 512 environment configurations and text instruction pairs.

## 5.2 Training details

To implement our auto-regressive transformer model, we used the GPT-2 (Radford et al., 2019) model architecture and tokenizer from the HuggingFace Library (Wolf et al., 2020). In all experiments, we used a reduced architecture with 4 layers, 2 attention heads and a dropout of 0.1, and we tested two hidden size configurations for the transformer: 256 and 512. We train our model on the generated dataset using AdamW optimizer (Loshchilov & Hutter, 2017) with a learning rate of 1e-4.

At every epoch, we perform 5000 model updates during which we sample 128 sub-trajectories of size $K$ from the dataset. Instructions, actions and captions are tokenized using the pre-trained GPT-2 tokenizer, and embedded with a common embedding layer. Observations are encoded using a 3-layer convolutional network, then flattened with a linear layer. To predict the action or caption token, we use a linear layer on the transformer hidden state followed by softmax.

## 5.3 Comparison to caption-free Baselines

We first compare our unified policy to the baseline that does not use subgoals. This baseline is trained using the exact same process as the caption-based policy, except that the captions were removed from the dataset, and only actions remain. This baseline can therefore been seen as reasoning-free model, which at train time, only sees the instruction, observations and the actions. We performed all experiment with 3 random seeds on the three babyAI levels, and show results in Table 1. We also reported the numbers from the imitation learning baseline (**BabyAI-Ori-1M**) of the original BabyAI paper (Chevalier-Boisvert et al., 2018). This model was trained on expert demonstrations with language-grounded instructions, in the same setup as our baseline, with the only difference being the dataset size: all the models from the original BabyAI imitation learning baseline were trained on 1M trajectories, vs. twice as less for our baseline on the **Synth** task, and 10 times less on **PutNextLocal**. We see that both baselines perform similarly on all three levels.

Concerning the caption-based model, we observe that it performs similarly to the baselines on **PutNextLocal-100k**, the simplest task, but that the baseline is slightly better on **Synth-500k**. On the harder **BossLevel-1M** task however, our model that leverages captions outperforms baselines, showing on average +10% success rate. These results can be explained by the nature of the tasks. Indeed BossLevel requires long-term reasoning and sequential planning, while a good exploration behaviour and a decent understanding of the instruction is sufficient for solving most of the tasks from the Synth environment. This result is strengthen by the graphs shown in Fig. 4, which compare our method and the baseline for varying training dataset sizes, sub-trajectory length $K$ and hidden size. We see that our method consistently outperforms the caption-free baseline for all three parameters, highlighting the efficiency of leveraging captioned subgoals for challenging sequential tasks.

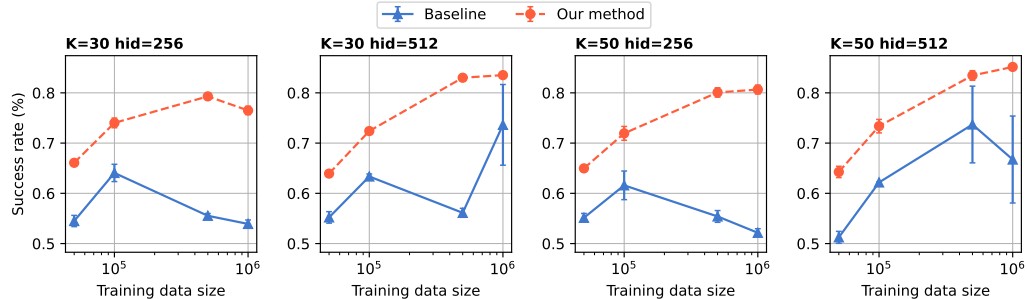

Figure 4: Success rate on BossLevel for varying amount of training samples in 4 different settings (varying subtrajectory length $K$ and the model hidden size). Our method consistently outperforms the caption-free baseline. We repeat each experiment 3 times and plot standard error.

Table 2: Comparison of encoding strategies for the caption-free baseline and our method.

|  | Encoding | PutNextLocal-100k | BossLevel-1M |
|---|---|---|---|
| **Baseline** | position | 98.9 ±0.1 | 57.8 ±3.9 |
|  | timestep/sequence | 99.5 ±0.2 | 73.6 ±8.0 |
| **Our Method** | position | 99.0 ±0.3 | 78.1 ±0.6 |
|  | timestep | - | 79.6 ±1.1 |
|  | sequence | 99.6 ±0.1 | 85.2 ±0.5 |

### 5.4 Importance of positional encoding

Finally, we investigate the role of positional encoding in our model and the baseline. This encoding embeds the relative position of elements stored in the batch fed as input to the transformer. It plays an important role as its allows the model to attend to the right elements. We compare three different types of encodings: **(i)** the *position* encoding, which is the absolute position in the sub-sequence fed to the model, as in the original transformer implementation, **(ii)** the *timestep* encoding that encodes the actual environment timestep of the corresponding observation and action, and **(iii)** the *sequence* encoding, which corresponds to the index in the modified sequence, a default one as described in Section 4.1. Note that for the caption-free baseline, **(ii)** and **(iii)** are the same since, in the absence of captions, the modified sequence is the same as the original trajectory.

We report results in Table 2, for **PutNextLocal-100k** and **BossLevel-1M** tasks. We see that, on the simpler task, the switching to position encoding only slightly degrades the performance ($< 1\%$) for the baseline and our method. On BossLevel however, the performance of position encoding is much worse than the sequence one: -16% for the baseline, and -7% for our method. This confirms the intuition that sequence encoding is necessary for tasks that require knowing where the agent is in the episode for predicting the right actions and subgoals. We also note that timestep encoding, while being slightly better than position encoding, is much worse than sequence ones. A potential reason for that is the fact that actions and captions from the same timesteps have the same encoding, and it might make it hard for the model to disentangle both elements.

## 6 Discussion

We proposed a unified policy that is capable of switching between reasoning with language and selecting the next action to take. When trained on offline data that contains subgoal descriptions, we achieved consistent improvements over the baseline that ignores this language reasoning. One possible explanation for this improvement is that language reasoning splits complex tasks into smaller chunks that are easier for the model to learn. This promising result suggests the possibility of improving RL training efficiency with language reasoning.

While language reasoning was restricted to subgoal generation in this paper, the same mechanism can be used for other types reasoning, such as common-sense reasoning and making inferences. In future work, we aim to extend our method to more complex environments such as MineCraft using the MineDojo dataset that contains video captions.

## Acknowledgements

Karteek Alahari is supported in part by the ANR grant AVENUE (ANR-18-CE23-0011).

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
