# OpenReview forum: "Think Before You Act: Unified Policy for Interleaving Language Reasoning with Actions"
_ICLR.cc/2023/Workshop/RRL — RRL 2023 Poster_

### Official Review · Reviewer_HDjr · 2023-02-26
**Unifying Language and Actions into a Single Policy**

**Rating:** 2
**Confidence:** 3

**Review:**

This paper proposes unifying language reasoning with actions into a single policy. The authors utilize captions in offline training data and propose using captions as well as actions and observations to construct a model that is trained to predict both actions and text captions. The paper presents results showing that this model trained to predict both captions and actions performs better in the BabyAI environment. The paper's strengths include suggesting an interesting modification to existing language-conditioned RL methods and positive initial results. However, the paper's experiments section is not very significant. While initial results appear positive, it would be helpful to see this method tested on more environments. The novelty of this paper is also questionable. The authors do not provide technical analysis as to the differences between their work and the work proposed by Li et al. that they reference in their Related Works section. The method section lacks technical details and theoretical justification that is arguably necessary when studying their proposed architecture, both in sections 4.1 and 4.2.

---

### Official Review · Reviewer_W2ev · 2023-02-27
**Novel and interesting ideas**

**Rating:** 2
**Confidence:** 3

**Review:**

This work presents a transformer-based policy for offline goal-conditioned task learning. It is a novel and interesting way that the model integrates expert instruction (caption) with the state-action sequences for task trajectory generation.

Strengths：
1) This work provides an unifying model that combines NLP and RL-sequencee generation for offline-goal-conditioned imitation learning.  As the transformer shows remarkable performance on NLP tasks, this paper presents a natual and intuitive method for AI to learn expert language-conditioned demonstrations.
2) Unlike other language-conditioned RLs learn the representation of language goals, this work provides an end-to-end way that integarates language goal into the control policy.

Weakness:
1) This work only presents the comparison results between caption-augmented GPT and non-augmented GPT. It would be nice to see the comparisons with other offline RLs.
2) Does this caption-augmented offline method also work when the offline dataset contains suboptimal and random trajectories?
3) Is the caption transformer policy robust? Can it handle stochastic environments and disturbances?
4) The test environment task is relatively simple. Seeing the following work with more complicated/continuous tasks would be nice.

Detailed questions:
1) Table 1: It seems that the caption augmented version only significantly contributes Boss-level if 1M trajectories are given. Why each difficult level has different data-size?
2) From Fig.4, it indicates that caption-augmented transformer outperforms the baseline on Boss-level difficulty. What about the other two-levels? Can we still get the conclusion?
3) Does the caption transformer has the task decomposition ability? For example, the agent is in the same environment but is given a new task consists multiple sub-tasks from different goal-conditoned trajectories.

Topics:
This work belongs to offline RL/IL, which can be considered as a sub-topic of Reincarnating RL.